# Real-World Experience with Cemiplimab Treatment for Advanced Cutaneous Squamous Cell Carcinoma—A Retrospective Single-Center Study

**DOI:** 10.3390/jcm12185966

**Published:** 2023-09-14

**Authors:** Daniella Kuzmanovszki, Norbert Kiss, Béla Tóth, Veronika Tóth, József Szakonyi, Kende Lőrincz, Judit Hársing, Enikő Kuroli, Eleonóra Imrédi, Tünde Kerner, Mihály Patyánik, Norbert M. Wikonkál, Ákos Szabó, Valentin Brodszky, Fanni Rencz, Péter Holló

**Affiliations:** 1Department of Dermatology, Venereology and Dermatooncology, Faculty of Medicine, Semmelweis University, H-1085 Budapest, Hungary; kiss.norbert@med.semmelweis-univ.hu (N.K.); toth.bela@med.semmelweis-univ.hu (B.T.); toth.veronika@med.semmelweis-univ.hu (V.T.); szakonyi.jozsef@med.semmelweis-univ.hu (J.S.); lorincz.kende@med.semmelweis-univ.hu (K.L.); harsing.judit@med.semmelweis-univ.hu (J.H.); kuroli.eniko@med.semmelweis-univ.hu (E.K.); nardaine_imredi.eleonora@med.semmelweis-univ.hu (E.I.); kerner.tunde_zsuzsanna@med.semmelweis-univ.hu (T.K.); wikonkal.norbert@med.semmelweis-univ.hu (N.M.W.); hollo.peter@med.semmelweis-univ.hu (P.H.); 2Uzsoki Street Hospital, Practice Hospital of the Faculty of General Medicine, Semmelweis University, H-1145 Budapest, Hungary; patyanik@uzsoki.hu; 3Central Hospital of Northern Pest-Military Hospital, H-1139 Budapest, Hungary; 4Department of Health Policy, Corvinus University of Budapest, H-1093 Budapest, Hungary; akos.szabo@uni-corvinus.hu (Á.S.); valentin.brodszky@uni-corvinus.hu (V.B.); fanni.rencz@uni-corvinus.hu (F.R.); 5Károly Rácz Doctoral School of Clinical Medicine, Semmelweis University, H-1085 Budapest, Hungary

**Keywords:** cemiplimab, cutaneous squamous cell carcinoma, PD-1 inhibitor, immune checkpoint inhibitor, adverse event, immunotherapy, immunocompromised patients

## Abstract

Background: The systemic treatment of advanced cutaneous squamous cell carcinoma (cSCC) has seen significant developments in recent years. The anti-PD1 inhibitor cemiplimab has demonstrated efficacy in clinical trials, but real-world data are still limited. Here, we aimed to evaluate the efficacy and the safety of cemiplimab in a real-world clinical setting. Methods: A retrospective analysis was carried out for all patients who received at least two doses of cemiplimab at our department between February 2020 and January 2023. Progression-free survival (PFS), overall survival (OS), the objective response rate (ORR), the disease control rate (DCR) and adverse events (AEs) were evaluated. Results: Twenty-five patients were included with a median age of 78 (65–82) years. The median treatment duration was 48 (16–72) weeks. Five (20%) patients were immunocompromised. Sixteen patients (64%) developed AEs, including 36% serious AEs (SAEs) of grade ≥ 3. Six patients (24%) were withdrawn from treatment due to the occurrence of AEs. Among the 25 patients, 52% showed an objective response (3 complete and 10 partial responses), 76% had controlled disease and 24% experienced progression. Among the five immunocompromised patients, the ORR was 60%, while the DCR was 80%. Conclusions: This retrospective real-world study revealed that locally advanced or metastatic cSCC could be effectively treated with cemiplimab even in elderly, polymorbid and immunocompromised patients.

## 1. Introduction

Cutaneous squamous cell carcinoma (cSCC) is the second most common type of skin cancer after basal cell carcinoma, with an increasing incidence [1]. There are several risk factors for cSCC, such as UV exposure, immunosuppression, elderly age, chronic inflammation, wounds and genodermatoses [1,2,3]. Approximately one-fourth of all cases are observed among immunocompromised patients, in particular in solid organ transplant recipients (OTRs) [4,5,6].

Most primary cSCC cases are indolent tumors showing a very good prognosis, with 5-year cure rates of more than 90% and metastasis occurring in only 5% of the patients [1,2]. Surgery with or without radiotherapy remains the gold-standard treatment for cSCC. Advanced cSCC includes locally advanced (la-cSCC) and metastatic cutaneous squamous cell carcinoma (m-cSCC), which are not amenable to curative surgery or radiotherapy [1,7].

Patients with advanced cSCC show a poor prognosis when they are administered conventional systemic therapy such as platinum-based cytotoxic chemotherapy or epidermal growth factor receptor (EGFR) inhibitors. While advanced cSCC might respond to these treatments, in most cases the response is not durable [8,9].

The development of the immune checkpoint inhibitor (ICI) cemiplimab has led to a breakthrough in the treatment of la-cSCC. Cemiplimab is a recombinant IgG4 human monoclonal antibody anti-PD-1 inhibitor approved for the treatment of advanced cSCC by the FDA in 2018 and the EMA in 2019 after encouraging results from phase II trials [1,7,8,10,11].

Here, we aimed to evaluate the efficacy and the safety of cemiplimab in patients with advanced cSCC in a real-world clinical setting. 

## 2. Materials and Methods

### 2.1. Study Population

The study population consisted of patients with unresectable locally advanced or/and metastatic cSCC treated with the PD-1 inhibitor cemiplimab at the Department of Dermatology, Venereology and Dermatooncology, Faculty of Medicine, Semmelweis University, between 1 February 2020 and 31 January 2023. Inclusion criteria were the following: the presence of la-cSCC or/and m-cSCC that was treated with the PD-1 ICI cemiplimab during the study period for at least two cycles of the standard dosing of cemiplimab (350 mg every 3 weeks administered intravenously). Our study included solid organ transplant recipients and patients with poor performance status (ECOG more than 1) and an immunocompromised state. There was no upper age limit.

### 2.2. Clinical Data Collection

Data were collected retrospectively from our electronic medical records software (e-MedSolution version 2023/Q2/5, T-Systems Hungary, Budapest, Hungary), as follows: demographic data of the patients, disease stage at first presentation, laboratory results and baseline characteristics before the initiation of cemiplimab therapy. Data cutoff was set to 31 January 2023, giving a minimum follow-up of three months. Staging was performed based on the 8th edition of the TNM classification for invasive cSCC by the UICC and AJCC (2017) [12]. Performance status of the patients was assessed according to the Eastern Cooperative Oncology Group (ECOG) [13]. Data were collected on the overall survival (OS), progression-free survival (PFS), objective response rate (ORR), disease control rate (DCR) to cemiplimab and treatment-related adverse events (AEs). Therapeutic effect was classified as partial response (PR), complete response (CR), progressive disease (PD) or stable disease (SD). The disease control rate (DCR) was defined as the rate of patients with CR, PR and SD. The categorization of the response was carried out by the attending dermato-oncologist treating the patients based on immune-related Response Evaluation Criteria in Solid Tumors (iRECIST). Treatment-related adverse events were characterized using the Common Terminology Criteria for Adverse Events (CTCAE) version 5.0.

Cemiplimab treatment was administered until disease progression, demise of the patient, unacceptable toxicity or if the treating dermato-oncologist decided to withdraw the drug for other reasons.

### 2.3. Statistical Analysis

Real-world OS (rwOS) was calculated using the date of initiation of cemiplimab therapy and date of the demise of the patient or last follow-up. Real-world progression-free survival (rwPFS) was based on the initiation of the treatment and the date of disease progression or demise of the patient, or in patients without progression the date of the last follow-up. We analyzed the tumor responses (CR, PR, SD and PD) as well as the ORR and DCR. RwOS and rwPFS times were estimated using the Kaplan–Meier survival analysis method and differences between subgroups were compared by log-rank test. Fisher’s exact test or Mann–Whitney U test was utilized for descriptive analysis, as applicable; *p* < 0.05 was considered statistically significant with the confidence interval set to 95%. All statistical analyses were performed using the IBM SPSS Statistics for Windows software, version 25.0. (IBM Corp, Armonk, New York, NJ, USA). 

## 3. Results

### 3.1. Patient Characteristics 

A total of twenty-five patients were included: the median age was 78 years (range: 65–82.5) and 68% were men (Table 1). Seventeen patients (68%) were older than 70 years. Nine (36%) patients had a T stage of T4a. Two patients (8%) had an unknown T stage (Tx).

Twenty patients (80%) had locally advanced cSCC and five patients (20%) had distant metastases, which were all pulmonary metastases.

The primary tumor localization of cSCC was the head and neck area in 17 (68%) patients, upper or lower extremities in 3 (12%) and on the trunk in 5 (20%) patients. Nineteen patients (73%) were ECOG 0 status, five (20%) ECOG 1 and one (4%) ECOG 2.

At the beginning of the treatment, 14 (56%) patients had normal kidney function (≥60 GFR ml/L/m^2^L) and 11 (44%) patients had a GFR < 60. Seven patients (28%) had marked anemia (hemoglobin level: 80–100 g/L). 

Five (20%) patients had hypertension, two (8%) had ischemic heart disease, four (16%) had both diseases together, one (45) had diabetes mellitus, two (8%) had both diabetes and hypertension and six (24%) had hypertension, ischemic heart disease and diabetes mellitus simultaneously. 

Thirteen patients (52%) had another malignancy: most of them (eight patients, 32%) had basocellular carcinoma, three (12%) had malignant melanoma and four (16%) had a lymphoproliferative disorder. 

Of the 25 patients, 5 (20%) were immunocompromised, 4 (16%) had had chronic lymphocytic leukemia (CLL) and 1 was a kidney transplant recipient. 

### 3.2. Treatment Characteristics

Twenty-two patients (88%) received cemiplimab as first-line treatment and three (12%) as second-line treatment. For the patients who received cemiplimab as second-line therapy, the histological findings showed mixed basosquamous carcinoma before therapy; therefore, they received first-line treatment with vismodegib with no improvement, until repeated histological examination confirmed cSCC.

Twelve patients (48%) received radiation therapies prior or simultaneously: five (20%) to the locally advanced tumor, five (5%) to the lymph nodes and two (8%) to both regions. 

### 3.3. Treatment Efficacy

The ORR was 52% in the whole population. Among the immunocompromised subgroup it was 60%, in the la-cSCC group it was 55% and in the m-cSCC group it was 40% (Figure 1).

The DCR was 76% in the whole investigated population (Table 2), while the immunocompromised patients reached higher rates: 80% in the la-cSCC group and 60% in the m-cSCC group (Table 3). 

The differences in demographics and clinical characteristics of the responders (*n* = 13), cSCC patients who achieved an objective response, and non-responders (*n* = 12), patients who achieved SD or PD as the best response, were compared. The grade of AE data for one patient was not available (NA). The association of the examined parameters with survival outcomes was also investigated.

At the time of data analysis (January 2023), treatment with cemiplimab was ongoing in ten (40%) patients and the median duration of treatment was 48 weeks. Fifteen patients (60%) discontinued cemiplimab due to disease progression in six cases (24%), toxicity in six cases (24%) and death due to unrelated causes (not related either to tumor or to therapy) in two patients (8%).

The median OS and PFS were not reached yet. Based on the high ORR data, the PFS and OS values were close to each other, and thus we used the mean OS value as a basis for the survival analysis.

The mean OS was 85.06 weeks (Figure 2). We noted that for those patients who were older than 70 years, the median survival was as high (87.05 weeks) as younger patients (<70 years, 83.75 weeks).

Patients with M1 status with distant metastases had a shorter median survival (42.37 weeks) compared to the patients with locally advanced (91.38 weeks) and regional in-transit and/or lymph node metastases (65.88 weeks).

Immunocompromised patients achieved similar survival as the whole investigated population.

The presence of any grade of an irAE meant a worse outcome, and patients who had no AE during the cemiplimab treatment showed a longer median survival (76.43 weeks vs. 45.29 weeks *p =* 0.542). The irradiation also did not improve the survival outcomes.

We noticed shorter median survival in the patients with low kidney function values (GFR < 60 mL/L/m^2^L).

### 3.4. Toxicity

We detected 34 cases of adverse events: 16 (64%) patients experienced at least one treatment-mediated immune-related adverse event (irAE); 9 (36%) patients had grade 3–4 adverse events; and in 11 cases (10 patients, 40%) the AEs led to hospitalization. Six (24%) patients had a serious irAE that led to discontinuation of the treatment. There were no fatal adverse events (grade 5).

The most common adverse events were thyroiditis (occurring in 24% of the patients), nephritis (16%), anemia (16%), colitis (12%) and pancreatitis (12%) (Table 4). Adverse events of grade 3 or higher that occurred in two patients were pneumonitis and nephritis, in one patient was severe colitis, neutropenia, pancreatitis and myositis, and in one patient was pneumothorax (PTX). Adverse events that were assessed by investigators to be related to the treatment are shown in Table 4.

In the subgroup of the five immunocompromised patients, three had irAEs and were immunocompromised, and none of them discontinued cemiplimab on the grounds of toxicity.

The kidney transplant recipient patient’s kidney function decreased following cemiplimab treatment. The initial GFR was 66 mL/L/m2L, which decreased to 30 mL/L/m^2^L after 10 months of therapy. However, the PD-1 inhibitor treatment led to partial remission of the cSCC. The decrease in kidney function values may also be due to the simultaneous bisphosphonate infusion from the 6th cycle introduced due to bone involvement. In this case, the use of cemiplimab did not lead to rejection of the transplanted organ.

## 4. Discussion

Here, we presented the results of a single-center retrospective study of patients with advanced unresectable cSCC treated with anti-PD-1 cemiplimab.

This cohort consisted of 25 patients with known unfavorable characteristics, such as being elderly individuals (≥70 years old), being immunocompromised, with kidney failure or with a polymorbid condition. The majority of these patients could not be enrolled in the appropriate phase III trials as they would fail to meet inclusion and exclusion criteria [8,11,14,15]. In one published clinical trial (EMPOWER-cSCC-1), patients with active autoimmune disease, lymphoproliferative disorders and other infectious comorbidities as well as patients with performance status higher than 1 were excluded [11].

We compared the results of our real-world retrospective analysis with the efficacy and safety data obtained in clinical trials [8,11,14,15]. 

In our study, the ORR was 52%, compared with 44% (95% CI 32–55) in the phase II trials on anti-PD-1 antibodies [8]. We observed a similar rate of complete response (CR) (12% vs. 11.3%) to that reported in the updated analysis of the EMPOWER-cSCC-1 trial. However, in our study, the rate of partial response (PR) was found to be higher (40% vs. 33.9%) [11]. During a longer follow-up period, it is possible that some of the patients that only achieved a PR could show further improvement and reach a CR [11,14,15]. 

In addition, in our study, for 88% of the patients, cemiplimab was administered as the first-line systemic treatment, versus 66% in the phase II trial [15].

IrAEs developing as a result of cemiplimab treatment often require treatment by systemic glucocorticoids or other immunosuppressant medication [16]. Here, we report severe irAEs (grade 3 or 4) in 36% of the investigated patients, while mild irAEs (grade 1 or 2) developed in 60%. In our cohort, the most common mild irAE was thyroiditis (24%), and the most frequent grade 3–4 irAEs were pneumonitis and nephritis. In a phase II trial, grade 3–4 treatment-emergent adverse events occurred in 44% of patients, with the most common being hypertension (8%) and pneumonia (5%) [8]. The most common adverse events regardless of attribution were fatigue (27.0%) and diarrhea (23.5%) in another phase II trial [15]. Advanced cSCC patients are frequently polymorbid elderly individuals, as also seen in our study, with a median age of 78 years and a high mean number of comorbidities. Although the risk of severe irAEs is not elevated in elderly patients with cancer [17], even milder irAEs could result in serious complications in this frail patient population.

Previous real-world studies have reported optimal outcomes of anti-PD1 treatments in advanced cSCC. A German observational retrospective multicenter study reported on 46 patients suffering from advanced cSCC treated with checkpoint inhibitors (cemiplimab, nivolumab or pembrolizumab). In this analysis, they found an ORR of 58.7%, with 15.2% of patients exhibiting CRs and a DCR of 80.4% [18]. In addition, a study from France reporting on the cemiplimab treatment of 61 patients reported an ORR of 50.4% and a DCR of 59.6% [19]. In an Italian retroprospective cohort study where elderly individuals were recruited, 23 responses (76.7%) with 9 CRs (30%) were achieved., An overall response was reported in four of five immunosuppressed patients included in this study [20].

Our study showed higher ORRs and DCRs in the patients with la-cSSC compared to patients with distant metastases (ORR: 55% vs. 40%; DCR: 80% vs. 60%), and the mean survival time proved to be longer. In an observational retrospective multicentric study from Germany, they found that distant metastases did not affect the response when compared with locally advanced disease. In this study, they could identify two predictive factors. Those individuals who had high lactate dehydrogenase (LDH) serum levels at the start of treatment and those patients that had their primary tumor located on the lower extremity displayed poorer outcomes. In their study, only 10% of the patients discontinued treatment due to toxicity, and thus good treatment tolerability was concluded [18]. A further recent real-world study revealed better response to immunotherapy in patients with head and neck cSSC. On the other hand, cSCC of genital areas had the poorest treatment outcomes [21]. In our cohort, among the 25 investigated patients, only 3 (12%) had cSCC on extremities and 5 (20%) on the trunk, with the survival being shorter in patients with primary tumors on the head and neck region. At the beginning of the cemiplimab therapy, all of our patients had normal LDH serum levels. 

Solid-organ transplantation patients under immunosuppression show a 65 to 250 times higher risk of cSCC when compared to healthy individuals [4,6]. The majority of cSCC patients harbor hypermutated tumors due to the mutagenic activity of sun exposure [4,5,22,23]. The combination of a high number of tumor mutations and the higher incidence in immunocompromised patients makes cSCC a good target for immune therapy [8,14].

The findings of our real-world study reveal that immunocompromised patients under cemiplimab therapy, in contrast to their exclusion from most clinical trials, show favorable outcomes and survival data. However, previous studies revealed that anti-PD-1 increased the risk of graft rejection, especially in liver transplant recipients [19,22]. An ongoing clinical trial involving kidney transplantation patients with advanced cSCC is investigating the efficacy and safety outcomes of cemiplimab in combination with everolimus or sirolimus and prednisone therapy [19]. 

Upon the evaluation of the subgroup of patients with lymphoproliferative disorders, a high ORR was found. This is in accordance with previous studies, where immunotherapy was effective in this patient population [24].

In our study, >60 GFR and >100 g/L hemoglobin at the beginning of cemiplimab therapy were associated with a minor increase in survival rates. However, no difference was observed with respect to median age, sex, T and N stage, irradiation, the presence of adverse events or any further disorder or other tumor. 

The limitations of this study include the small number of patients, the heterogenous patient population, a monocentric setting, the absence of long-term data and the retrospective nature. 

## 5. Conclusions

In our real-life study, cemiplimab showed a high efficacy, with an acceptable safety profile similar to those in clinical trials with carefully selected patients. Moreover, it was also proven effective in polymorbid and elderly patients and in individuals with immunocompromised status.

## Figures and Tables

**Figure 1 jcm-12-05966-f001:**
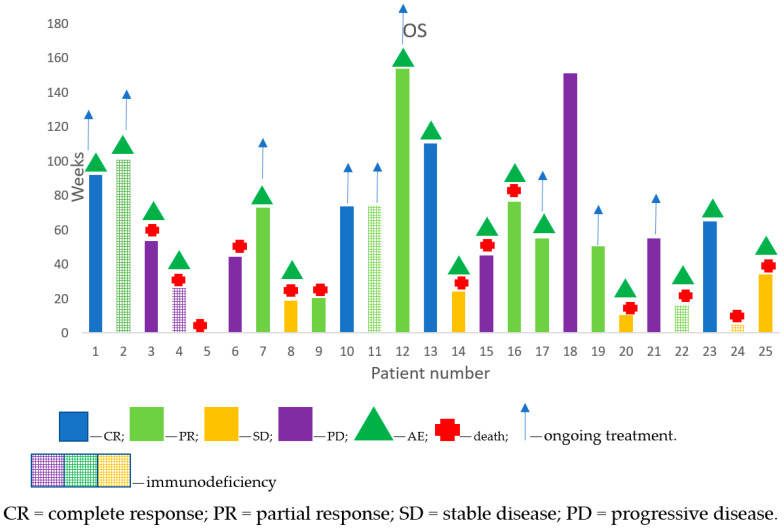
Treatment outcomes of the investigated patients.

**Figure 2 jcm-12-05966-f002:**
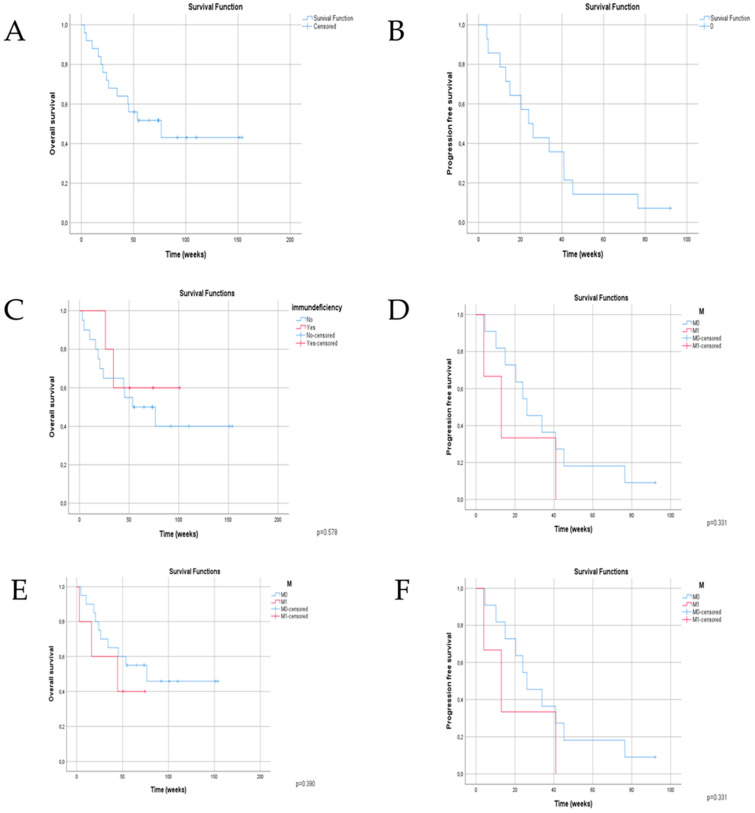
Kaplan—Meier curves for overall survival (**A**)**,** progression-free survival (**B**), OS in patients with immunodeficiency (**C**), PFS in patients with immunodeficiency (**D**), OS in patients with distant metastasis (**E**) and PFS in patients with distant metastasis (**F**).

**Table 1 jcm-12-05966-t001:** Patients’ demographic and clinical characteristics.

	Total Sample * (N = 25)	Responders * (N = 13)	Non-Responders * (N = 12)	*p*-Value **
Age (years)	78.00 (65.00–82.50)	78.00 (66.50–80.00)	79.50 (57.75–85.00)	0.564
≥70 years	17 (68.00%)	9 (69.23%)	8 (66.67%)	1.000
<70 years	8 (32.00%)	4 (30.77%)	4 (33.33%)	
Received doses (piece)	12.00 (4.50–19.50)	19.00 (12.00–20.50)	5.00 (3.00–11.75)	<0.001
Duration of treatment (weeks)	48.00 (16.43–72.43)	68.43 (51.22–82.85)	20.79 (9.04–40.97)	<0.001
Gender				
male	17 (68.00%)	9 (52.94%)	8 (47.06%)	1.000
female	8 (32.00%)	4 (50.00%)	4 (50.00%)	
T				
Tx	2 (8.00%)	1 (50.00%)	1 (50.00%)	0.796
T1	1 (4.00%)	0 (0.00%)	1 (100.00%)	
T2	6 (24.00%)	3 (50.00%)	3 (50.00%)	
T3	7 (28.00%)	3 (42.86%)	4 (57.14%)	
T4a	9 (36.00%)	6 (66.67%)	3 (33.33%)	
N				
N0	13 (52.00%)	6 (46.15%)	7 (53.85%)	0.755
N2a	1 (4.00%)	0 (0.00%)	1 (100.00%)	
N2b	1 (4.00%)	1 (100.00%)	0 (0.00%)	
N2c	4 (16.00%)	3 (75.00%)	1 (25.00%)	
N3a	3 (12.00%)	2 (66.67%)	1 (33.33%)	
N3b	3 (12.00%)	1 (33.33%)	2 (66.67%)	
M				
M0	20 (80.00%)	11 (55.00%)	9 (45.00%)	0.645
M1	5 (20.00%)	2 (40.00%)	3 (60.00%)	
Localization of the primary tumor				
Head/neck	17 (68.00%)	9 (52.94%)	8 (47.05%)	
Limb	3 (12%)	2 (66.66%)	1 (33.33%)	1.000
Trunk	5 (20.00%)	2 (40.00%)	3 (60.00%)	
Site of metastases				
Locally advanced	14 (56.00%)	8 (57.14%)	6 (42.86%)	0.868
Lymphonodular, in transit	6 (24.00%)	3 (50.00%)	3 (50.00%)	
Distant	5 (20.00%)	2 (40.00%)	3 (60.00%)	
Line of treatment				
First	22 (88.00%)	12 (54.55%)	10 (45.45%)	0.593
Second	3 (12.00%)	1 (33.33%)	2 (66.67%)	
Hemoglobin (g/L)				
Normal	15 (60.00%)	9 (60.00%)	6 (40.00%)	0.431
80–100	7 (28.00%)	2 (28.57%)	5 (71.43%)	
101–120	3 (12.00%)	2 (66.67%)	1 (33.33%)	
Creatinine (µmol/L)				
0	17 (68.00%)	10 (58.82%)	7 (41.18%)	0.509
1	7 (28.00%)	3 (42.86%)	4 (57.14%)	
2	1 (4.00%)	0 (0.00%)	1 (100.00%)	
GFR (ml/L/m^2^L)	64.00 (58.00–90.00)	64.00 (60.00–90.00)	64.00 (51.25–88.25)	0.568
≥60	14 (56.00%)	8 (57.14%)	6 (42.86%)	0.695
<60	11 (44.00%)	5 (45.45%)	6 (54.55%)	
ECOG				
0	19 (76.00%)	11 (57.89%)	8 (42.11%)	0.467
1	5 (20.00%)	2 (40.00%)	3 (60.00%)	
2	1 (4.00%)	0 (0.00%)	1 (100.00%)	
Irradiation				
No	13 (52.00%)	6 (46.15%)	7 (53.85%)	0.695
Yes	12 (48%)	7 (58.33%)	5 (41.67%)	
Site of irradiation				
No	13 (52.00%)	6 (46.15%)	7 (53.85%)	0.753
T	5 (20.00%)	3 (60.00%)	2 (40.00%)	
N	5 (20.00%)	2 (40.00%)	3 (60.00%)	
Both	2 (8.00%)	2 (100.00%)	0 (0.00%)	
AE				
No	9 (36.00%)	7 (77.78%)	2 (22.22%)	0.097
Yes	16 (64.00%)	6 (37.50%)	10 (62.50%)	
Grade of AE (missing = 1)			(missing = 1)	
Gr 1–2	9 (37.50%)	3 (33.33%)	6 (66.67%)	0.157
Gr: 3–4	6 (25.00%)	3 (50.00%)	3 (50.00%)	
0	9 (37.50%)	7 (77.78%)	2 (22.22%)	
AE				
0	9 (37.50%)	7 (70.00%)	2 (22.22%)	0.308
1	6 (24.00%)	3 (50.00%)	3 (50.00%)	
>1	9 (36.00%)	3 (33.33%)	6 (66.67%)	
Other disorders				
0	5 (20.00%)	2 (40.00%)	3 (60.00%)	0.442
DM	1 (4.00%)	1 (100.00%)	0 (0.00%)	
HT	5 (20.00%)	3 (60.00%)	2 (40.00%)	
IHD	2 (8.00%)	2 (100.00%)	0 (0.00%)	
HT + IHD	4 (16.00%)	3 (75.00%)	1 (25.00%)	
All 3	6 (24.00%)	2 (33.33%)	4 (66.67%)	
HT + DM	2 (8.00%)	0 (0.00%)	2 (100.00%)	
Other tumor				
No	12 (48.00%)	8 (66.67%)	4 (33.33%)	0.238
Yes	13 (52.00%)	5 (38.46%)	8 (61.54%)	
Immunodeficiency				
No	20 (80.00%)	10 (50.00%)	10 (50.00%)	1.000
Yes	5 (20.00%)	3 (60.00%)	2 (40.00%)	
CLL				
No	21 (84.00%)	11 (52.38%)	10 (47.62%)	1.000
Yes	4 (16.00%)	2 (50.00%)	2 (50.00%)	

AE = adverse event; CLL = chronic lymphocytic leukemia; ECOG = Eastern Cooperative Oncology Group Performance Status; GFR = glomerular filtration rate; PFS = progression-free survival; DM = diabetes mellitus; HT = hypertension; IHD = ischemic heart disease. * The non-responder (stable and progressive disease) and responder (partial and complete responses) subgroups were created based on the responders’ therapy responses. ** Mann–Whitney U test or Fisher’s exact test. A *p*-value < 0.05 was considered statistically significant.

**Table 2 jcm-12-05966-t002:** Survival analysis of the patients.

		N	Mean or Median Survival (Weeks)	LCI	UCI	*p*-Values (Log-Rank)
Total sample (median)		25	*85.06*	*59.01*	*111.10*	-
Age (years)	≥70	17	*87.05*	*56.78*	*117.31*	0.882
	<70	8	*83.75*	*38.22*	*129.29*	
Gender	male	17	*90.13*	*59.18*	*121.09*	0.773
	female	8	*73.50*	*33.00*	*114.00*	
Localization of the primary tumor	head/neck	17	*62.86*	*44.24*	*81.47*	0.833
	trunk	5	*93.23*	*31.18*	*155.28*	
	limb	3	104.10	24.46	183.74	
T	Tx	2	*50.22*	*0.00*	*101.60*	0.211
	T1	1	*16.14*	*16.14*	*16.14*	
	T2	6	*71.90*	*38.85*	*104.95*	
	T3	7	*79.63*	*33.06*	*126.21*	
	T4a	9	*108.70*	*66.90*	*150.50*	
N	N0	13	-	-	-	0.742
	N2a	1	-	-	-	
	N2b	1	-	-	-	
	N2c	4	-	-	-	
	N3a	3	-	-	-	
	N3b	3	-	-	-	
M	M0	20	*89.38*	*61.01*	*117.75*	0.390
	M1	5	*42.37*	*16.75*	*67.99*	
Site of metastases	Locally advanced	14	*91.38*	*58.57*	*124.19*	0.675
	In transit	6	*65.88*	*29.76*	*101.99*	
	Distant	5	*42.37*	*16.75*	*67.99*	
GFR (mL/L/m^2^L)	≥60	14	*94.81*	*59.40*	*130.22*	0.265
	<60	11	*56.76*	*31.99*	*81.54*	
Site of irradiation	No	13	*93.26*	*56.93*	*129.58*	0.964
	T	5	*63.22*	*34.08*	*92.35*	
	N	5	*48.63*	*28.70*	*68.55*	
	Both	2	*47.07*	*10.15*	*83.99*	
Irradiation	No	13	*93.26*	*56.93*	*129.58*	0.643
	Yes	12	*58.11*	*38.50*	*77.71*	
AE	No	9	**76.43**	**2.96**	**149.90**	0.542
	Yes	16	**45.29**	**27.65**	**62.93**	
Grade of AE (missing = 1)	Gr 1–2	9	*87.03*	*47.34*	*126.72*	0.806
	Gr: 3–4	6	*53.38*	*15.35*	*91.42*	
	0	9	*70.03*	*43.48*	*96.58*	
AE	0	10	*95.36*	*55.04*	*135.68*	0.197
	1	6	*72.60*	*49.54*	*95.66*	
	>1	9	*62.08*	*19.04*	*105.12*	
Other tumor	No	12	*96.38*	*58.49*	*134.28*	0.304
	Yes	13	*55.00*	*34.54*	*75.46*	
Responders’ therapy responses	PD + SD	13	*35.94*	*14.41*	*57.47*	**<0.001**
	PR + CR	12	*129.30*	*98.97*	*159.63*	

LCI = lower bound of the 95% confidence interval; PD + SD = progressive disease and stable disease; PR + CR = partial response and complete response; UCI = upper bound of the 95% confidence interval. A *p*-value < 0.05 was considered statistically significant. The italic values refer to mean and bold values refer to median overall survival weeks.

**Table 3 jcm-12-05966-t003:** ORR and DCR.

Responses	Total (N = 25)	Immunodeficiency (N = 5)	la-cSCC (N = 20)	m-cSCC (N = 5)
CR	3 (12%)	0	3 (15%)	0
PR	10 (40%)	3 (60%)	8 (40%)	2 (40%)
SD	6 (24%)	1 (20%)	5 (25%)	1 (20%)
PD	6 (24%)	1 (20%)	4 (20%)	2 (40%)
ORR *	13 (52%)	3 (60%)	11 (55%)	2 (40%)
DCR **	19 (76%)	4 (80%)	16 (80%)	3 (60%)

CR = complete response; PR = partial response; SD = stable disease; PD = progressive disease; * ORR = objective response rate; DCR = disease control rate; ORR = CR + PR, ** DCR = CR + PR + SD.

**Table 4 jcm-12-05966-t004:** Adverse events.

AE (Type)	AE (all): 34 No. of Patients: 16 (64%)	AE Grade 1–2: 23 No. of Patients: 14 (60%)	AE Grade 3–4: 9 No. of Patients: 9 (36%)	AE Led to Hospitalization: 11 No. of Patients: 10 (40%)	AE Led to Permanent Discontinuation of Treatment: 8 No. of Patients: 6 (24%)
anemia	4 (16)	4 (16)	0	0	0
neutropenia	1 (4)	0	1 (4)	1 (4)	1 (4)
eosinophilia	1 (4)	1 (4)	0	0	0
fatigue	2 (8)	2 (8)	0	0	0
thyroiditis	6 (24)	6 (24)	0	0	0
^1^ IDDM	2 (8)	1 (4)	0	1 (4)	0
pancreatitis	3 (12)	2 (8)	1 (4)	1 (4)	1 (4)
pneumonitis	2 (8)	0	2 (8)	2 (8)	2 (8)
colitis	3 (12)	2 (8)	1 (4)	1 (4)	1 (4)
myositis	1 (4)	0	1 (4)	1 (4)	1 (4)
nephritis	4 (16)	2 (8)	2 (8)	2 (8)	2 (8)
skin reaction	2 (8)	2 (8)	0	1 (4)	0
infection	2 (8)	2 (8)	0	0	0
^2^ PTX	1 (4)	0	1 (4)	1 (4)	0

^1^ IDDM = insulin-dependent diabetes mellitus, ^2^ PTX = pneumothorax.

## Data Availability

All relevant data are available upon reasonable request from the corresponding author.

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
