# Peer review of "Real-World Experience with Cemiplimab Treatment for Advanced Cutaneous Squamous Cell Carcinoma—A Retrospective Single-Center Study"

_jcm, 2023, doi:10.3390/jcm12185966_

Round 1

Reviewer 1 Report

The authors present an interesting real life experience on SCC patients treated with cemiplimab. Despite the limitation related to the sample size (i.e., only 25 patients), the study is well designed and in line with previously reported cohorts. Some minor revisions may be needed to improve the overall quality of the draft. For instance:

1) Could the authors provide more details on the "high risk" features of the analysed SCC cases, as defined by the EADO-EADV guidelines (e.g. G3, perineural invasion, localisation on ear/lip/temple)?

2) Have the authors investigated if such high-risk features may have an impact on response/survival outcomes? 

2) The survival analysis curves have poor image resolution. May the authors replace them with higher quality images?

Minor english revision is required

Author Response

The authors present an interesting real life experience on SCC patients treated with cemiplimab. Despite the limitation related to the sample size (i.e., only 25 patients), the study is well designed and in line with previously reported cohorts. Some minor revisions may be needed to improve the overall quality of the draft. For instance:

1) Could the authors provide more details on the "high risk" features of the analysed SCC cases, as defined by the EADO-EADV guidelines (e.g. G3, perineural invasion, localisation on ear/lip/temple)?

We thank the reviewer for his positive feedback on our manuscript.

In our study, the histological findings of the primary tumor showed poor differentiation in all cases. Regarding the risk factors, in two cases the primary tumor was larger than 6 cm. In two cases, the histology described perineural invasion. In three patients, the primer cSCC developed on the eyelid, in two cases in the temporal region, and in one case in the ear.

2) Have the authors investigated if such high-risk features may have an impact on response/survival outcomes? 

Thank you for this suggestion. However, as very few cases affected the high-risk groups separately, we were not able to investigate the impact of high-risk features on response or survival.

2) The survival analysis curves have poor image resolution. May the authors replace them with higher quality images?

Thank you for pointing out that a lower resolution figure was uploaded. We have replaced it with a high-quality image.

Reviewer 2 Report

Line 39: advise changing "and wounds and" to ",wounds and"

Line 43: advise changing "dissemination" to "metastasis"

Line 68: what do you mean under "poor performance status in an immunocompromised state"? This sentence conflicts with Line 124 and Table 1 data. 

Line 109: Correct sentence to "The primary tumor localization of cSCC was the head and neck area in 17 109 (68%) patients, upper or lower extremities in three (12%) and on the strain in five (20%) patients."

Line 123: You state that 5 patients were immunocompromised which is in conflict with Line 68. CLL can be considered an immunocompomised state, please explain or reconcile those two statements. 

Line 124: Correct "leucaemia" to "leukemia". Can you confirm, did ALL patients had CLL? It conflicts with Table 1 data. Correct "have chronic ..." to "had chronic.."

Table 1 and 2. What to do you mean under "strain" for location of tumor? Should it be "trunk"?

Line 202: Can you comment on the kidney-transplant patient's kidney function/survival after treatment? Using PD-1 inhibitors may lead to loss of transplanted organ and it is important to collect data about outcome of treatment in regard of transplanted organ survival. Paragraph starting with Line 206 addresses this issue, however no comment is provided on specific renal transplant patient in the study. 

Line 250. Place period at the end of the sentence. 

See above. Minor corrections needed. 

Author Response

Line 39: advise changing "and wounds and" to ",wounds and"

Corrected.

Line 43: advise changing "dissemination" to "metastasis"

Corrected.

Line 68: what do you mean under "poor performance status in an immunocompromised state"? This sentence conflicts with Line 124 and Table 1 data. 

We have clarified poor performance status as “ECOG more than 1” in the manuscript.

Line 109: Correct sentence to "The primary tumor localization of cSCC was the head and neck area in 17 109 (68%) patients, upper or lower extremities in three (12%) and on the strain in five (20%) patients."

Corrected.

Line 123: You state that 5 patients were immunocompromised which is in conflict with Line 68. CLL can be considered an immunocompomised state, please explain or reconcile those two statements. 

In this study, we also included immunocompromised patients, such as those with CLL.

Line 124: Correct "leucaemia" to "leukemia". Can you confirm, did ALL patients had CLL? It conflicts with Table 1 data. Correct "have chronic ..." to "had chronic.."

Four of our patients had CLL, we have revised this part to avoid confusion.

Table 1 and 2. What to do you mean under "strain" for location of tumor? Should it be "trunk"?

Yes, we have replaced to “trunk” at all occurrences.

Line 202: Can you comment on the kidney-transplant patient's kidney function/survival after treatment? Using PD-1 inhibitors may lead to loss of transplanted organ and it is important to collect data about outcome of treatment in regard of transplanted organ survival. Paragraph starting with Line 206 addresses this issue, however no comment is provided on specific renal transplant patient in the study. 

The following text was inserted to the suggested part: “The kidney-transplant recipient patient's kidney function decreased following cemiplimab treatment. The initial GFR was 66 ml/L/m²L, that decreased to 30 ml/L/m²L after 10 months of therapy. However, the PD-1 inhibitor treatment lead to partial remission of the cSCC. The decrease of kidney function values may also be due to the simultaneous bisphosphonate infusion from the 6th cycle introduced due to the bone involvement. In this case, the use of cemiplimab did not lead to rejection of transplanted organ.”

Line 250. Place period at the end of the sentence. 

Corrected.